# Antibiotic interactions shape short-term evolution of resistance in *E. faecalis*

Ziah Dean[1], Jeff Maltas[1], Kevin B. Wood[1,2]*

**1** Department of Biophysics, University of Michigan, Ann Arbor, Michigan, United States of America,
**2** Department of Physics, University of Michigan, Ann Arbor, Michigan, United States of America

* kbwood@umich.edu

**Data Availability Statement:** We have posted all data to Data Dryad (https://datadryad.org/) where it is freely available without restriction.

**Funding:** This work is supported by the National Science Foundation (NSF No. 1553028 to KW), the

## Abstract

Antibiotic combinations are increasingly used to combat bacterial infections. Multidrug therapies are a particularly important treatment option for *E. faecalis*, an opportunistic pathogen that contributes to high-inoculum infections such as infective endocarditis. While numerous synergistic drug combinations for *E. faecalis* have been identified, much less is known about how different combinations impact the rate of resistance evolution. In this work, we use high-throughput laboratory evolution experiments to quantify adaptation in growth rate and drug resistance of *E. faecalis* exposed to drug combinations exhibiting different classes of interactions, ranging from synergistic to suppressive. We identify a wide range of evolutionary behavior, including both increased and decreased rates of growth adaptation, depending on the specific interplay between drug interaction and drug resistance profiles. For example, selection in a dual β-lactam combination leads to accelerated growth adaptation compared to selection with the individual drugs, even though the resulting resistance profiles are nearly identical. On the other hand, populations evolved in an aminoglycoside and β-lactam combination exhibit decreased growth adaptation and resistant profiles that depend on the specific drug concentrations. We show that the main qualitative features of these evolutionary trajectories can be explained by simple rescaling arguments that correspond to geometric transformations of the two-drug growth response surfaces measured in ancestral cells. The analysis also reveals multiple examples where resistance profiles selected by drug combinations are nearly growth-optimized along a contour connecting profiles selected by the component drugs. Our results highlight trade-offs between drug interactions and resistance profiles during the evolution of multi-drug resistance and emphasize evolutionary benefits and disadvantages of particular drug pairs targeting enterococci.

## Author summary

Antibiotics are increasingly used in combinations to combat difficult bacterial infections, including those caused by the opportunistic pathogen *E. faecalis*. Some pairs of drugs exhibit particularly strong inhibitory effects when used simultaneously, as each drug magnifies the effects of the other, a phenomenon known as synergy. Other drug combinations may counteract one another, an effect called antagonism. These antibiotic "interactions"

National Institutes of Health (NIH No.
1R35GM124875-01 to KW), and the Hartwell
Foundation for Biomedical Research (to KW). The
funders had no role in study design, data collection
and analysis, decision to publish, or preparation of
the manuscript.

**Competing interests:** The authors have declared
that no competing interests exist.

form the basis of powerful antimicrobial therapies, but they may also impact the emergence of drug resistance in surprising ways. In this work, we studied the evolution of antibiotic resistance using laboratory evolution of *E. faecalis* populations exposed to different two-drug combinations. We found that cells adapt more rapidly to certain drug combinations, while other antibiotic pairs markedly slow resistance. The modulated adaptation results from the interplay of drug interactions and the evolved resistance to each individual drug. Despite a wide range of observed evolutionary behavior, we find that our results can be intuitively understood using simple rescaling arguments that describe how resistance mutations impact the effective concentration of each drug. These findings highlight trade-offs between drug interactions and resistance profiles during the evolution of multi-drug resistance and emphasize evolutionary benefits and disadvantages of particular drug pairs targeting enterococci.

## Introduction

The rapid rise of antibiotic resistance poses a growing threat to public health [1, 2]. The discovery of new antimicrobial agents is a long and difficult process, underscoring the need for new approaches that optimize the use of currently available drugs. In recent years, significant efforts have been devoted to designing evolutionarily sound strategies that balance short-term drug efficacy with the long-term potential to develop resistance. These approaches describe a number of different factors that could modulate resistance evolution, including interactions between bacterial cells [3–8], synergy with the immune system [9], spatial heterogeneity [10–15], epistasis between resistance mutations [16, 17], precise temporal scheduling [18–21], and statistical correlations between resistance profiles for different drugs [22–31].

Drug combinations are an especially promising and widely used strategy for slowing resistance [32], and there has been significant work devoted to identifying and predicting the effects of different drug cocktails [33–39]. From a clinical perspective, synergistic interactions–where the combined effect of the drugs is greater than expected based on the effects of the drugs alone [33]–have long been considered desirable because they provide strong antimicrobial effects at reduced concentrations. By contrast, drug pairs that interact antagonistically–effectively weakening one another in combination–have been traditionally avoided. Work over the last decade has challenged this conventional wisdom by demonstrating that synergistic interactions have a potentially serious drawback: they may accelerate the evolution of resistance [40–42]. Similarly, antagonistic interactions can slow or even reverse the evolution of resistance [43]. These results indicate that drug interactions underlie a natural trade-off between short-term efficacy and long-term evolutionary potential [44]. In addition, recent work has shown that cross-resistance (or collateral sensitivity) between drugs in a combination may also significantly modulate resistance evolution [26, 27, 45]. As a whole, these studies show that drug interactions and collateral effects may combine in complex ways to influence evolution of resistance in multi-drug environments.

Antibiotic combinations are often the most effective therapies for enterococcal infections, which lead to significant morbidity and mortality [46–51]. *E. faecalis* is among the most commonly isolated enterococcal species and underlies a host of human infections, including infective endocarditis, infections of the urinary tract or blood stream, and infections related to surgical devices and medical implants. Multiple combination therapies have been proposed or are currently in use for *E. faecalis* infections [46, 52, 53]. While synergistic combinations are

the standard–particularly for high inoculum infections–relatively little is known about how different combinations affect the potential for, and rate of, resistance evolution.

To address these questions, we use large scale laboratory evolution to measure growth adaptation and phenotypic resistance in populations of *E. faecalis* exposed to four different two-drug combinations over multiple days. The drug pairs include several clinically relevant combinations–for example, two β-lactams or a β-lactam and an amimoglycoside–and exhibit a range of interactions, from synergistic to strongly antagonistic (suppressive). Our results reveal rich and at times surprising evolutionary behavior. In all cases, we find that different dosing combinations lead to significantly different rates of growth adaptation, even when the level of inhibition is constant. In some cases, differences in growth adaptation appear to be driven by selection for distinct cross-resistance profiles, while in other cases, strong interactions between the drugs lead to different adaption rates but highly similar profiles. Despite this apparent diversity, we show that qualitative features of these evolutionary trajectories can be understood using simple rescaling arguments that link resistance profiles in evolving populations to geometric transformations of the two-drug response surface in ancestral cells. Our results represent a quantitative case study in the evolution of multidrug resistance in an opportunistic pathogen and highlight both potential limitations and unappreciated evolutionary benefits of different drug combinations.

## Results

### Selection of antibiotic pairs with different interactions

We first set out to identify a set of two-drug combinations that include a range of interaction types: synergistic, antagonistic, and suppressive (i.e. strongly antagonistic so that the effect of the combination is less than that of one of the drugs alone). To do so, we measured the per capita growth rate of *E. faecalis* V583 populations in liquid cultures exposed to multiple drug pairs at 90-100 dosage combinations per pair. We estimated per capita growth rate in early exponential phase from optical density (OD) time series acquired using an automated microplate reader and plate stacker (Methods). The type of interaction for each drug pair is defined by the shape of the contours of constant growth ("isoboles") describing the growth response surface $g$ $(D_1, D_2)$, where $D_i$ is the concentration of drug $i$. Linear contours of constant growth represent additive (non-interacting) pairs–for example, the effect of one unit of drug 1 or drug 2 alone is the same as that of a combination with half a unit of each drug. Deviations from additivity include synergy (antagonism), which corresponds to contours with increased concavity (convexity), rendering an equal mixture of the two drugs more (less) effective than in the non-interacting scenario. While other metrics exist for quantifying drug interactions (see, for example, [33]), we choose this Loewe null model [54] because, as we will see, its simple geometric interpretation provides useful intuition for interpreting evolutionary trajectories [41, 43]. Based on these interaction measures and with an eye towards choosing clinically relevant combinations when possible, we decided to focus on 4 drug combinations: ceftriaxone (CRO) and ampicillin (AMP); ampicillin and streptomycin (STR); ceftriaxone and ciprofloxacin (CIP); and tigecycline (TGC) and ciprofloxacin. We describe these combinations in more detail below.

### Laboratory evolution across iso-inhibitory dosage combinations

Our goal was to compare evolutionary adaptation for different dosage combinations of each drug pair. The rate of adaptation is expected to depend heavily on the level of growth inhibition in the initial cultures, which sets the selection pressure favoring resistant mutants. To control for initial inhibition levels, we chose four dosage combinations for each drug pair—two

corresponding to single drug treatments and two to drug combinations—that lie (approximately) along a contour of constant inhibition (i.e. constant per capita growth rate). We then evolved 24 replicate populations in each dosage combination for 3-4 days (20-30 generations) with daily dilutions into fresh media and drug. The evolution experiments were performed in microwell plates, allowing us to measure time series of cell density (OD) for each population over the course of the adaptation. In addition, we characterized the phenotypic resistance of 6 randomly chosen populations per condition by measuring standard dose-response curves and estimating the half-maximal inhibitory concentration ($IC_{50}$) of each drug in the combination. Together, these measurements provide a detailed quantitative picture of both growth adaptation and changes in phenotypic resistance to each drug over the course of the evolution experiment.

## Dual β-lactam combination accelerates growth adaption but selects for similar resistance profiles as adaption to component drugs

Cell wall synthesis inhibitors, including β-lactams, are among the most frequently used antibiotics for *E. faecalis* infections [55]. While *E. faecalis* often exhibit sensitivity to aminopenicillins, such as ampicillin (AMP), they are intrinsically resistant to cephalosporins (e.g. ceftriaxone (CRO)). Despite the limited utility of ceftriaxone alone, it combines with ampicillin to form a powerful synergistic pair, making it an attractive option for *E. faecalis* harboring high-level aminoglycoside resistance. Dual β-lactam combinations like CRO-AMP have been particularly effective in treatment of endocarditis infections in the clinical setting [46, 52].

As expected, we found that the CRO-AMP combination is strongly synergistic in the ancestral V583 *E. faecalis* strain (Fig 1a, left panel). We selected four dosage combinations (labeled A-D) along the concave contour of constant inhibition and evolved replicate populations to each condition. Growth curves on day 0, the first day of evolution, show similar levels of inhibition for each combination, with growth initially increasing but later collapsing or plateauing. The growth curves on later days sometimes differ between replicates and between conditions (Fig 1a, right panels; S1 Fig). To quantify growth, we estimated the effective growth rate during an intermediate range of OD (0.1< OD<0.4) for each day and each condition using nonlinear least squares fitting to an exponential function (Fig 1b, left panels). For ancestor cells in the absence of drug, this regime corresponds to exponential growth (Fig 1a, right panels; S1 Fig) and this metric estimates the per capita growth rate. In cases where growth over this region is non-monotonic, this metric instead provides an effective measure of growth that decreases when population density declines, even if initial growth is rapid.

In all four conditions, growth increases significantly by day 2. Notably, late stage (days 2-3) growth rate is higher, on average, when both drugs are present (conditions B and C) than it is for the single-drug conditions (A and D). To further quantify these trends, we estimated the average adaptation rate for each population using linear regression for each growth rate time series (Fig 1b, right panel). While the 24 replicate populations in each condition show considerable variability–as expected, perhaps, for a stochastic evolutionary process–growth adaptation is significantly faster for the two combinations (B and C) than for the single drug treatments (A and D). These qualitative results do not depend sensitively on the specific OD window used to estimate growth (S2 and S3 Figs) and also hold when adaptation is estimated with a nonlinear function (S4 and S5 Figs). As an alternate way to visualize this adaptation, we also plotted the median growth curve (on final day of adaptation) calculated across all populations exposed to the same condition. The trends from this simple analysis are consistent with those from adaptation rate calculations–specifically, growth following adaptation to the drug combos is faster than growth following adaptation to the individual drugs (S6 Fig).

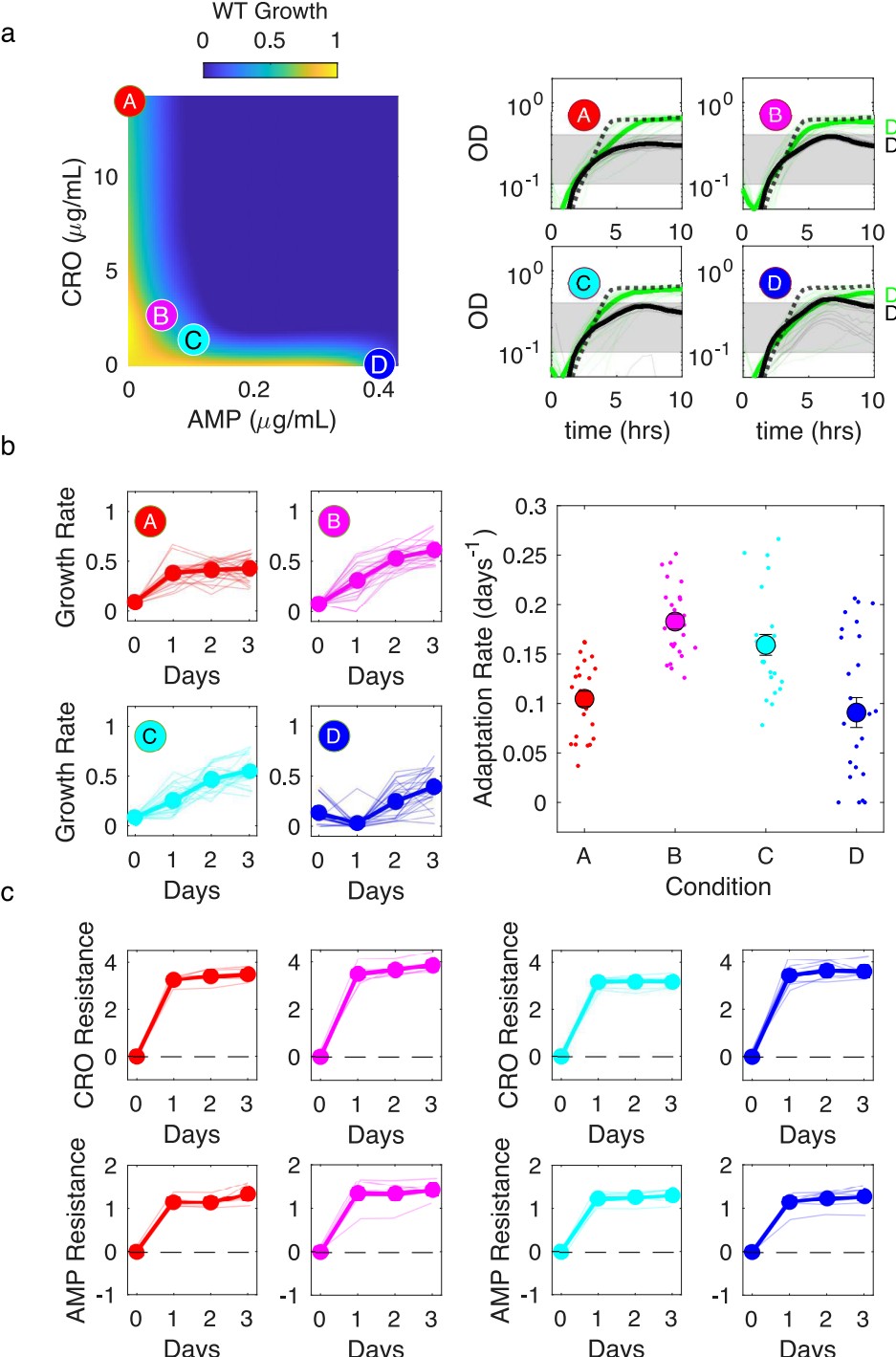

**Fig 1. Dual β-lactam combination accelerates growth adaption but selects for similar resistance levels to component drugs.** a) Left panel: per capita growth rate of ancestral populations as a function of ampicillin (AMP) and ceftriaxone (CRO) concentrations. Circles correspond to different selecting conditions along a contour of constant inhibition. Drug concentrations (AMP,CRO) are (0, 48) (red, A); (0.05, 2.6) (magenta, B); (0.10, 1.3) (cyan, C); and (0.43, 0) (blue, D). The latter point is shifted for visualization. Right panels: growth curves for the first (black) and last (green) days of evolution for each condition. Dashed lines are drug-free growth curves in ancestor strains. Shaded region is OD range over which per capita growth rate is estimated. b) Left panels: per capita growth rate over time for each condition. Right panel: average rate of growth adaptation over the course of the evolution. Adaptation rate in the single drug conditions (red and blue conditions, combined) is less than that in the combined drug conditions (magenta and cyan, combined), 1-sided t-test (unequal variance), $p < 10^{-4}$. Error bars are ± standard error of the

mean (SEM). c. Resistance to CRO (top panels) and AMP (bottom panels) over time for isolates from different conditions. Resistance is defined as the $log_2$-scaled fold change in $IC_{50}$ of the resistant isolate relative to ancestral cells (positive is increased resistance, negative is increased sensitivity). In all plots, light transparent lines correspond to individual populations and darker lines to the mean across populations.

One might naively expect that increased growth adaption in the drug combination indicates that populations in these conditions evolve higher levels of resistance to one or both drugs. To test this hypothesis, we used replicate dose-response measurements to estimate the half-maximal inhibitory concentration ($IC_{50}$) of both CRO and AMP in 6 randomly selected populations for each condition at the end of each day (see S7–S14 Figs for dose-response curves and S15 Fig for estimates of relative error). We then quantified resistance as the ($log_2$-scaled) fold change in $IC_{50}$ between the evolved and ancestral populations; positive values indicate increased resistance and negative values indicate increased sensitivity relative to the ancestral strain (Fig 1c). Note that resistance measured at the end of day 0 is considered to be resistance on day 1, as it is the expected resistance of the population at the start of day 1. Populations evolved under all four conditions exhibit similar patterns of phenotypic resistance, with $IC_{50}$'s to each drug rising rapidly and plateauing after 1-2 days. We note in passing that the growth of the population in condition D is near 0 on day 1, even though resistance to ampicillin is already seen in the $IC_{50}$ measurements; this peculiar finding may suggest that the population is heterogeneous, with resistant subpopulations comprising only a small fraction of the population on day 1. In all conditions, populations tend to show higher increases in CRO resistance than AMP resistance. These results are initially surprising because they indicate that mutants with nearly identical phenotypic resistance profiles nevertheless exhibit markedly different patterns of growth adaptation that depend on the specific dosage combination.

### Aminoglycoside/ β-lactam and β-lactam/fluoroquinolone combinations slow growth adaptation and select for resistant profiles distinct from those evolved to the component drugs

In addition to dual β-lactam therapies, combinations involving an aminoglycoside with a cell wall inhibiting antibiotic are commonly used for treating drug resistant *E. faecalis* [46]. In particular, the ampicillin and streptomycin (STR) combination has been used as a first line of treatment for *E. faecalis* infective endocarditis [46, 55]. Unfortunately, enterococci isolates are increasingly exhibiting high-level resistance to aminoglycosides, which has been shown to reduce the synergistic effect of the combination therapy [55]. While aminoglycoside resistance is a growing problem, the adaptation of aminoglycoside-resistant *E. faecalis* to combination therapies remains poorly understood.

Consistent with previous findings, we did not observe synergy between AMP and STR in the ancestral V583 strain, which exhibits considerable aminoglycoside resistance. In fact, the drug pair exhibits marked antagonism, as evidenced by the convex growth contours on the response surface (Fig 2a, left panel). Growth curves from populations evolved to four different conditions along a growth contour show considerable differences at day 2 between the single-drug conditions (conditions A and D) and the two-drug conditions (B and C, Fig 2a, right panel; S16 Fig), with single-drug populations reaching a higher growth rate (Fig 2b, left panels) and a dramatically increased rate of adaptation (Fig 2a, right panel; S17 Fig). At the end of the experiment (after day 2 of adaptation), populations grown in single drugs tend to show resistance to the selecting drug but little cross resistance to the other drug (Fig 2c, red and blue curves). Populations grown in a combination of both drugs show similar resistance profiles to those selected by the single drug that is dominant within the mixture. For example, the day 3

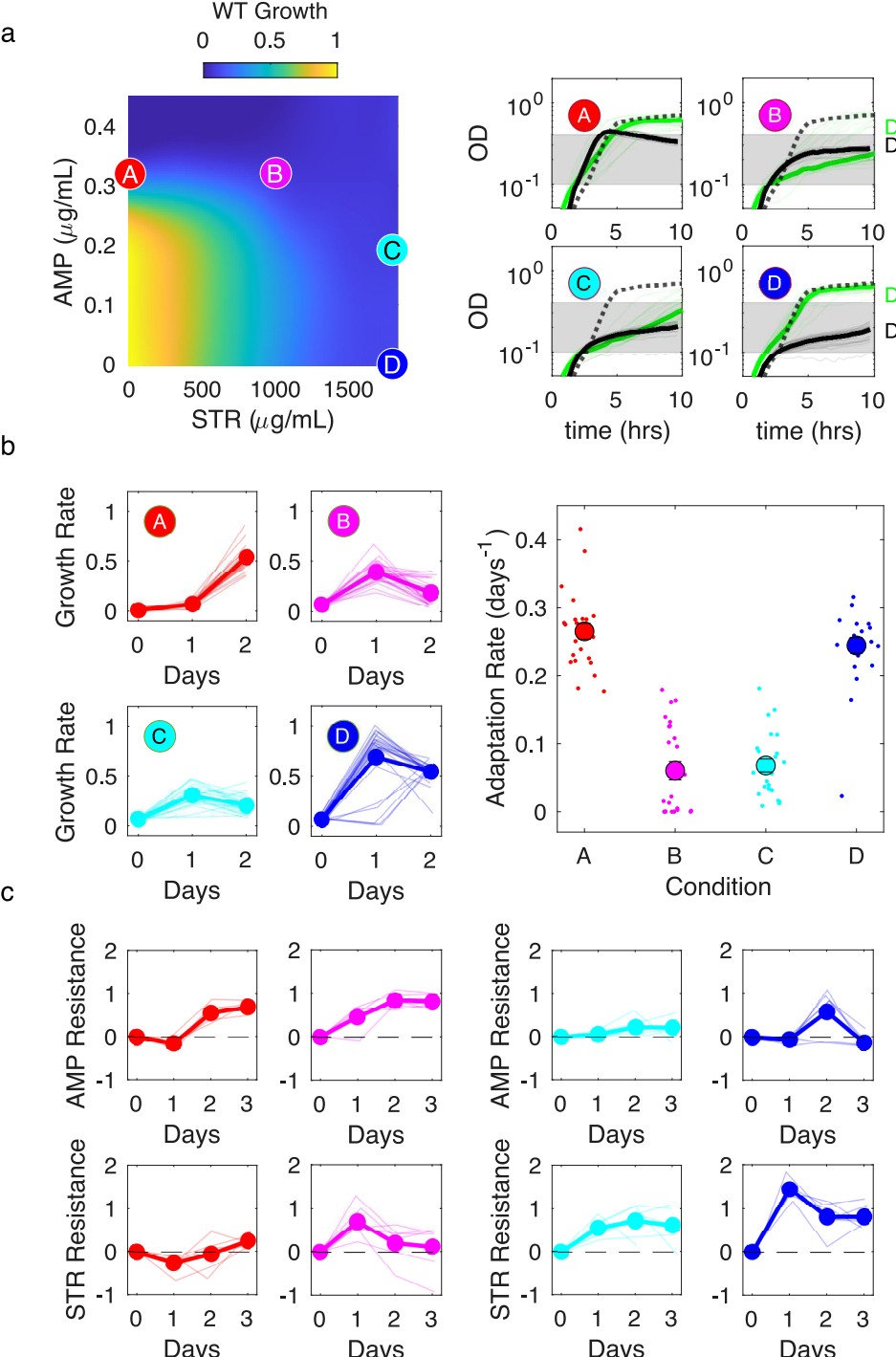

**Fig 2. Antagonistic β-lactam and aminoglycoside combination slows adaptation with little cross-resistance.** a) Left panel: per capita growth rate of ancestral populations as a function of ampicillin (AMP) and streptomycin (STR) concentrations. Circles correspond to different selecting conditions along a contour of constant inhibition. Drug concentrations (AMP,STR) are (0.3, 0) (red, A); (0.3, 1000) (magenta, B); (0.19, 1800) (cyan, C); and (0, 1800) (blue, D). Right panels: growth curves for the first (black) and last (green) days of evolution for each condition. b) Left panels: per capita growth rate over time for each condition. Dashed lines are drug-free growth curves in ancestor strains. Shaded region is OD range over which per capita growth rate is estimated. Right panel: average rate of growth adaptation over the course of the evolution. Adaptation rate in the single drug conditions (red and blue conditions, combined) is greater than that in the combined drug conditions (magenta and cyan, combined), 1-sided t-test (unequal variance), $p < 10^{-4}$. Error bars are ± standard error of the mean (SEM). c. Resistance to AMP (top panels)

and STR (bottom panels) over time for isolates from different conditions. Resistance is defined as the $\log_2$-scaled fold change in $IC_{50}$ of the resistant isolate relative to ancestral cells (positive is increased resistance, negative is increased sensitivity). In all plots, light transparent lines correspond to individual populations and darker lines to the mean across populations.

resistance profiles for condition B are similar to those from condition A (dominated by AMP), while those for condition C are similar to condition D (dominated by STR). It is interesting to note that the temporal dynamics (i.e. change in resistance over time for each drug) for all four conditions do show different qualitative trends, even when the final resistance profiles are similar (Fig 2c).

We observed qualitatively similar behavior in a combination of a β-lactam and fluoroquinolone (ciprofloxacin). Ciprofloxacin (CIP) is not typically used in the treatment of enterococci, though it has been used with β-lactams in the treatment of enterococcal endocarditis with high-level aminoglycoside resistance [56]. In vitro studies also demonstrate efficacy of ciprofloxacin in multiple combinations against *E. faecalis* biofilms [57]. We investigated resistance evolution in a combination of CIP with CRO, which is not a clinically used combination but exhibits less dramatic antagonism than the AMP-STR combination (Fig 3a, left panel), making it a potentially interesting proof-of-principle example of resistance involving fluoroquinolones.

As with the AMP-STR combination, we observe slowed growth adaptation in combinations of CRO-CIP relative to that in the component drugs alone (Fig 3b; S18 and S19 Figs). Adaptation to CRO alone leads to strong CRO resistance and slight CIP sensitivity (Fig 3c, red). On the other hand, adaptation to CIP alone leads to CIP resistance along with significant increases in CRO sensitivity (Fig 3c, blue). Populations evolved to combinations show different resistance profiles at different concentrations, though for both mixtures the collateral sensitivities are eliminated (Fig 3c, magenta and cyan).

## Tigecycline suppresses growth adaptation and eliminates evolution of fluoroquinolone resistance

The fourth combination we investigated was comprised of a protein synthesis inhibitor (tigecycline, TGC) and fluoroquinolone (CIP). TGC is a relatively new broad-spectrum antibiotic used for soft-tissue infections [58]; it also shows in-vitro synergy in combination with multiple antibiotics [59, 60]. We found that the TGC-CIP combination exhibits a particularly interesting type of interaction known as suppression (Fig 4a, left panel), where the combined effect of both drugs can be smaller than the effect of one drug (in this case, TGC) alone at the same concentration. Suppressive interactions between protein and DNA synthesis inhibitors in *E. coli* have been previously linked to sub-optimal regulation of ribosomal genes [61] as well as inverted selection for sensitive cells [43]. Because the growth contours in this case show nonmonotonic behavior, we performed evolution experiments in replicates of 8 for 11 different concentration combinations that fall along the isobole (Fig 4a; S20 and S21 Figs). We find that growth adaptation decreases approximately monotonically as TGC concentration is increased, eventually approaching a minimum as TGC eclipses a critical concentration $TGC_{crit} \approx 0.03$ μg/mL (Fig 4c). Furthermore, while we observed TGC resistance only in rare cases, populations adapted to TGC below the critical concentration show approximately constant levels of CIP resistance, while those at higher concentrations show essentially no CIP resistance (Fig 4d). It is particularly striking that populations evolved in conditions A (red) and I (light blue) exhibit such different evolutionary behavior, as both are exposed to nearly identical CIP concentrations and, by design, start at similar levels of inhibition. Yet evolution in condition A

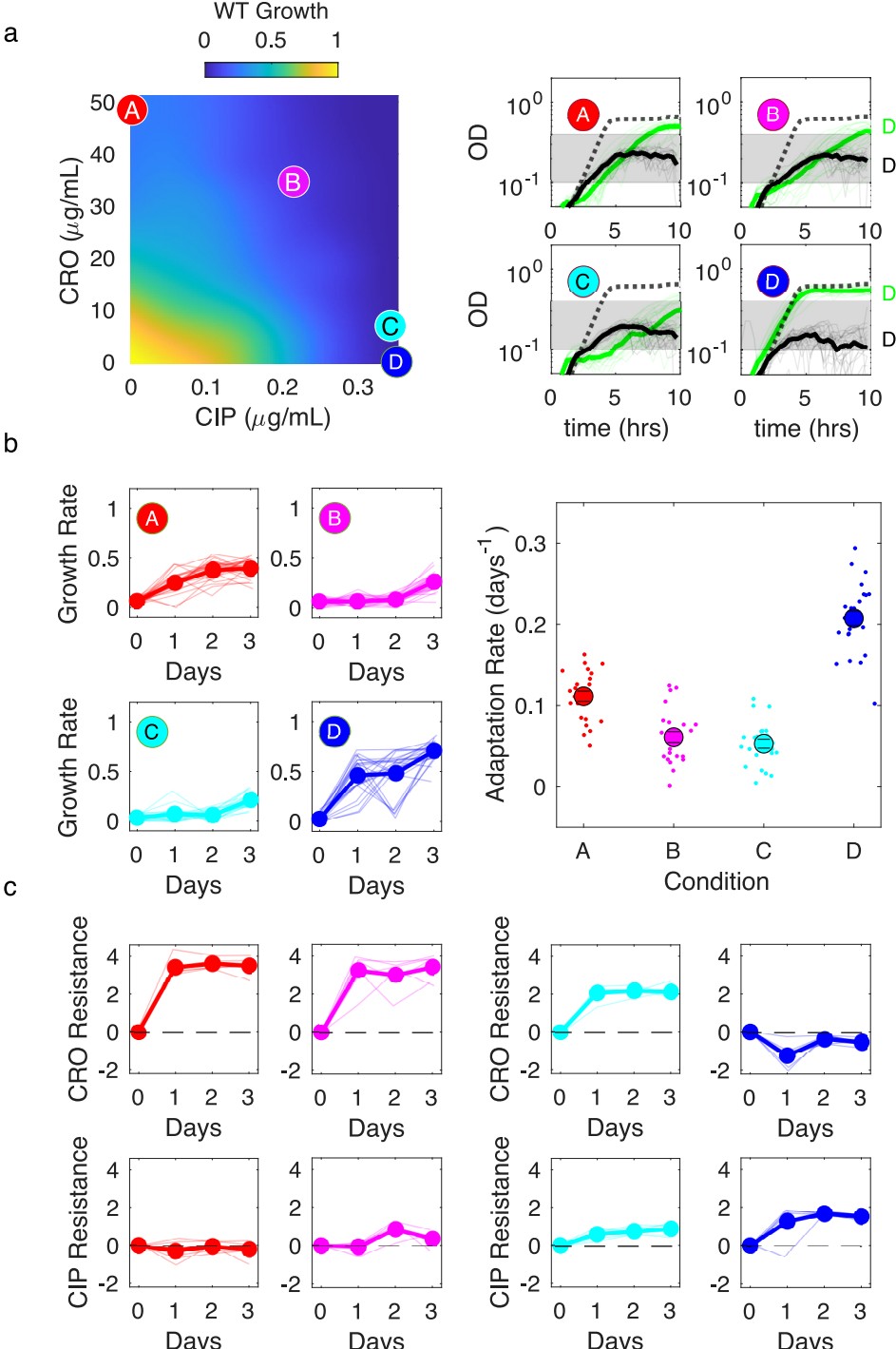

**Fig 3. Antagonistic β-lactam and fluoroquinolone combination slows adaptation and selects for populations lacking observed collateral sensitivity of single-drug isolates.** a) Left panel: per capita growth rate of ancestral populations as a function of ceftriaxone (CRO) and ciprofloxacin (CIP) concentrations. Circles correspond to different selecting conditions along a contour of constant inhibition. Drug concentrations (CRO, CIP) are (48.5, 0) (red, A); (34.7, 0.21) (magenta, B); (6.93, 0.34) (cyan, C); and (0, 0.46) (blue, D). The latter point is shifted for visualization. Right panels: growth curves for the first (black) and last (green) days of evolution for each condition. Dashed lines are drug-free growth curves in ancestor strains. Shaded region is OD range over which per capita growth rate is estimated. b) Left panels: per capita growth rate over time for each condition. Right panel: average rate of growth adaptation over the course of the evolution. Adaptation rate in the single drug conditions (red and blue conditions, combined) is greater than that in the combined drug conditions (magenta and cyan, combined), 1-sided t-test (unequal variance),

$p < 10^{-4}$. Error bars are ± standard error of the mean (SEM). c. Resistance to CRO (top panels) and CIP (bottom panels) over time for isolates from different conditions. Resistance is defined as the $\log_2$-scaled fold change in $IC_{50}$ of the resistant isolate relative to ancestral cells (positive is increased resistance, negative is increased sensitivity). In all plots, light transparent lines correspond to individual populations and darker lines to the mean across populations.

leads to fast growth adaptation and strong CIP resistance, while evolution in condition I (light blue) leads to little adaptation and no CIP resistance. In effect, the addition of TGC eliminates CIP resistance without modulating the overall efficacy of the (initial) combination.

## Geometric rescaling of ancestral growth surface explains condition-dependent growth adaptation when resistant profiles are unchanged

To interpret the observed evolutionary dynamics, we hypothesized that mutations conferring resistance modulate the effective drug concentration experienced by the population. In some cases–for example, resistance due to efflux pumps–this effective concentration change

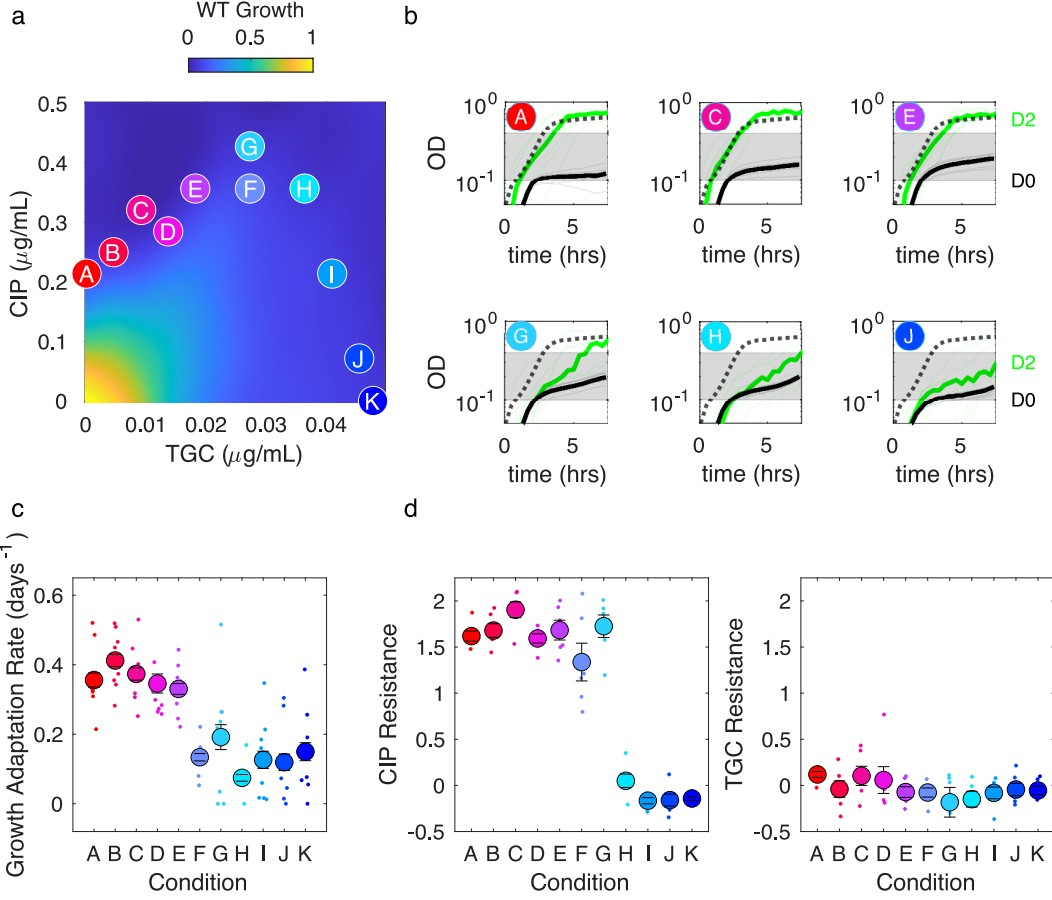

**Fig 4. Tigecycline eliminates fluoroquinolone resistance above a critical concentration.** a) Left panel: per capita growth rate of ancestral populations as a function of tigecycline (TGC) and ciprofloxacin (CIP) concentrations. Circles correspond to different selecting conditions along a contour of constant inhibition. b) growth curves for the first (black) and last (green) days of evolution for six of the 11 selecting conditions. c) Average rate of growth adaptation over the course of the evolution for each condition. d. Resistance to CIP (left) and TGC (right) for isolates on the final day of evolution. Resistance is defined as the $\log_2$-scaled fold change in $IC_{50}$ of the resistant isolate relative to ancestral cells (positive is increased resistance, negative is increased sensitivity). In all plots, light transparent lines or small points correspond to individual populations. Darker lines and larger circles represent means taken across populations. Error bars are ± standard error of the mean (SEM).

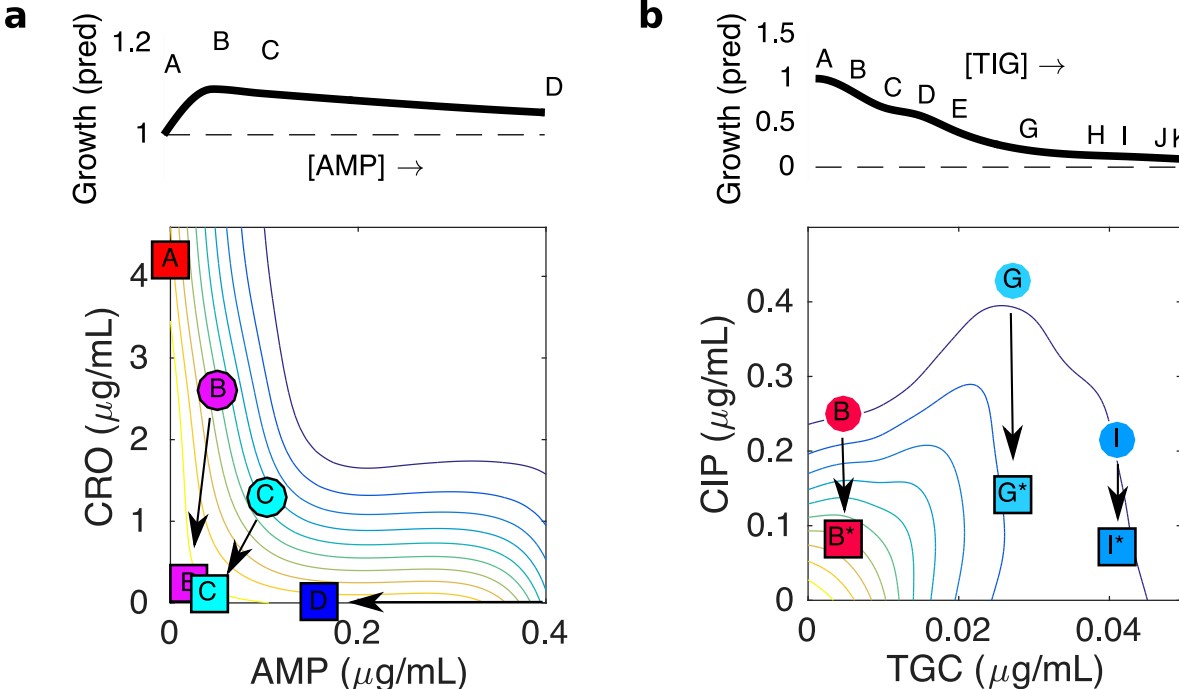

**Fig 5. Geometric rescaling of growth surface in ancestral strain explains growth differences between different conditions even when resistance profiles are identical.** Main panels: contour plots show growth in ancestral (WT) strain as function of drug concentrations for AMP-CRO (a, left) and TGC-CIP (b, right). Circles indicate selecting conditions (note that for visualization purposes, several selecting conditions are outside of the axis limits). Squares indicate effective concentrations achieved by rescaling true concentrations by the observed (mean) fold change in $IC_{50}$ for each drug. All arrows in a single panel correspond to the same rescaling: AMP→ AMP/2.5, CRO→ CRO/11.6 (panel a) and CIP→ CIP/3.0, TGC→ TCG (panel b). The rescaling factors correspond to cross resistance to AMP and CRO, with CRO resistance larger than AMP resistance (panel a) and to CIP resistance with no change in TGC resistance (panel b), which correspond to the mean values observed experimentally over all populations in AMP-CRO (see Fig 1) and the CIP-resistant populations in TGC-CIP (see Fig 4). Upper panels: predicted change in growth (relative to condition A) as one moves along the original contour, starting at condition A (one drug only). Predicted growth rate is calculated using 2d interpolation to estimate the growth along the rescaled contour on the ancestral growth surface.

corresponds to a genuine physical change in intracellular drug concentration. More generally, though, this hypothesis assumes that resistant cells growing in external drug concentration C behave similarly to wild type (drug-sensitive) cells experiencing a reduced effective concentration C' < C. Similar rescaling arguments were pioneered in [41, 43], where they were used to predict correlations between the rate of resistance evolution and the type of drug interaction.

Our results indicate that adaptation rates in CRO-AMP (Fig 1) and TGC-CIP (Fig 4) combinations can vary significantly across dosage combinations, even when resistance profiles are essentially unchanged. For example, populations adapted to CRO-AMP combinations show an average increase in $IC_{50}$ for CRO and AMP of about $2^{3.5} \approx 11.6$ fold and $2^{1.3} \approx 2.5$ fold, respectively. To understand how this level of resistance might be expected to impact growth, we rescaled the concentrations of CRO and AMP that lie along the contour of constant growth that passes through the four experimental dosage combinations (A-D). For each point on the contour, the concentrations of CRO and AMP are reduced by factors of 11.6 and 2.5, respectively. For example, the points corresponding to conditions A-D are mapped to the points shown in Fig 5a (squares). The new rescaled contour, which includes the rescaled locations of the original points A-D, does not in general correspond to a contour of constant growth on the original growth surface; therefore, growth of the adapted cells is expected to differ along the contour. More specifically, to predict growth of mutants selected along the original contour

of constant growth, one simply needs to find the rescaled contour, plot it atop the original (ancestral) growth surface, and read off the values of growth along that rescaled contour. In the CRO-AMP combination, this rescaling approach predicts that growth is slightly (approximately 10 percent or less) higher when the drugs are combined (e.g. conditions B and C) than when they are used individually (Fig 5a, top panel), in qualitative agreement with our experiments (Fig 1). More generally, the rescaling suggests that adaptation in all conditions should lead to dramatically increased growth, a consequence of the steep dose-dependence of the synergistic response surface.

Similarly, populations adapted in TGC-CIP show an increase in CIP $IC_{50}$ of approximately $2^{1.6} \approx 3$ fold, but only when adaptation occurs below a critical TGC concentration. If we apply the same rescaling approach–that is, we reduce the concentration of CIP by 3-fold for all points along the contour–we again get a series of new points that no longer fall on a single growth contour (Fig 5b, squares). Furthermore, the predicted growth for points on the new contour decreases monotonically with TGC concentration before plateauing near point G, near the critical concentration where experimental growth adaptation approaches its minimum value (Fig 5b). Intuitively, then, it becomes clear why selection for ciprofloxacin resistance is only favored below this critical concentration: for higher concentrations of TGC, the rescaled points fall very nearly on the same contour as the original point. That is, when the original contour becomes approximately vertical, rescaling the CIP concentration is no longer expected to increase growth (see, for example, point I).

## Resistance profiles selected in different AMP-STR and CRO-CIP combinations are (nearly) growth-optimized along contours connecting the profiles selected by component drugs

Rescaling arguments may also help us to understand why particular resistance profiles appear to be preferentially selected under different initial conditions, as we observed with AMP-STR and CRO-CIP. The resistance profiles on the final day of adaptation fall at different points in the two dimensional space describing resistance to each drug (Fig 6a and 6b, left panels). When profiles arising from adaptation to the same condition are averaged together, the resulting profiles (large circles) fall approximately on a smooth contour. We set out to determine how profiles at different points along these contours would be expected–based on rescaling–to impact growth in each of the four selecting conditions used experimentally.

To approximate these contours, we used a line segment (for AMP-STR) or a quadratic contour (for CRO-CIP) that connects conditions with extreme resistance. In both cases, the contours approximately run between red and blue points, corresponding to adaptation in the single drug conditions. To investigate how different resistance levels along these contours are expected to affect growth in each selecting condition, we rescaled the drug concentrations corresponding to each selecting condition by a range of rescaling factors that lie along the contours connecting their end points (labeled X and Y). As a result, each of the original selecting conditions (more specifically the points defined by the drug concentrations describing each condition) is mapped to a smooth curve in the two-drug concentration space. Points along that curve indicate how the original selecting concentrations are mapped to rescaled drug concentrations by the resistance profiles lying along the contour connecting X and Y. From that new curve, then, one can read off the corresponding growth rates from the ancestral growth surface, leading to predicted growth rates for mutants with any particular profile along the contour.

For each selecting condition, there is an optimal resistance profile (along the contour) that leads to the maximum possible expected growth. Not surprisingly, in the case of AMP-STR,

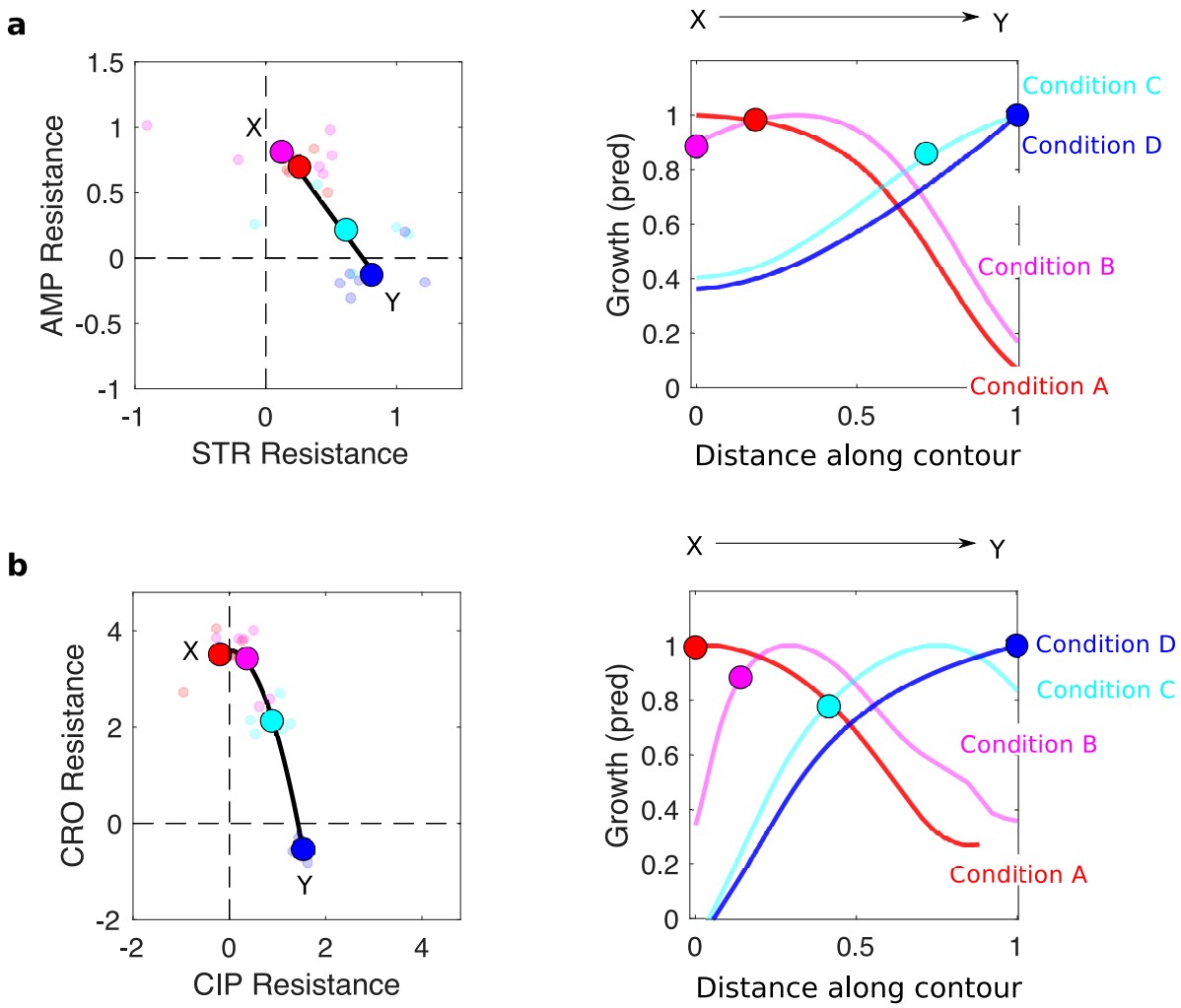

**Fig 6. Resistance profiles observed in different AMP-STR and CRO-CIP dosage combinations are (nearly) growth-optimized along the contour separating profiles selected by component drugs.** a: AMP-STR, b. CRO-CIP. Left panels: resistance profiles for individual populations (small circles) and the mean across populations (large filled circles) for each selecting condition. Color scheme is the same as in previous figures (i.e. red is condition A, magenta condition B, cyan condition C, and blue condition D). Solid black lines are linear (a) or quadratic (b) fits to the averaged resistance profiles, which create smooth contours in the 2-d resistance space. Right panels: each curve shows the predicted growth for populations with a range of resistance profiles (falling on the contours, left panels) grown at one specific selecting condition (A, red; B, magenta; C, cyan; D, blue). The horizontal axis corresponds to position along the contours (left panels). Filled circles correspond to the locations of experimentally observed (mean) resistance profiles.

maximum growth in condition A (AMP only) occurs at point X, where AMP resistance is highest. Similarly, maximum growth in conditions D (STR only) and C occur at point Y, which has the largest STR resistance. On the other hand, the optimal resistance profile for condition B lies just short of the midpoint on the line segment connecting X and Y (Fig 6a, right panel). Interestingly, the (mean) resistance profiles observed experimentally (circles) are predicted to give growth rates within approximately 15% of the optimal value.

In the case of the CRO-CIP combination, the optimal resistance profiles (along the contour) for conditions A (CRO only) and D (CIP only) lie at the endpoints X and Y, respectively (Fig 6b, right panel), which have the highest resistance levels to the component drugs. By contrast, the optimal profiles for conditions B (magenta) and C (cyan) fall at different points along the contour, reflecting trade-offs between resistance levels and collateral sensitivities to the

component drugs. Once again, the observed (mean) resistance profiles are predicted to give growth rates near the optimal values for each condition (particularly for conditions A, B, and D). Given that there is a finite number of genetic mutations possible under these short-term conditions, one would not expect that phenotypic profiles exist for all points along the contours. Nevertheless, these results suggest that for these two drug pairs, the resistant profiles selected in the combinations are expected to give growth benefits that are nearly optimal among all those possible along the contour.

## Discussion

Using laboratory evolution, we have shown that adaptation of *E. faecalis* populations to drug combinations can differ substantially from adaptation to the component drugs. While the evolutionary trajectory of any particular population is difficult to predict, the results as a whole point to simple trends that can be explained with rescaling arguments linking growth of adapted populations to growth of the ancestral population at properly rescaled drug dosages. These arguments show, for example, how identical resistance profiles yield different growth rates for different selecting conditions. The analysis also suggests that, in multiple cases, the profiles selected by different dosage combinations are very nearly growth-optimized along the contour connecting profiles selected by the individual component drugs. Given the inherent stochasticity of individual evolutionary trajectories, it is remarkable that such simple principles can be used to understand the mean behavior across large experimental ensembles.

It is important to point out several limitations to our study. First, our goal was not to investigate the specific molecular mechanisms involved in drug adaption, but instead to provide a quantitative picture of resistance evolution that does not require extensive molecular-level knowledge, which many times is not available. However, the richness of the observed phenotypes points to complex and potentially interesting genetic changes that can be partially resolved with modern sequencing technologies. For example, cross-resistance observed between ceftriaxone and ampicillin may be due to mutations in penicillin-binding proteins, which are common resistance determinants for both drugs [55]. We also note that our experiments were performed just below the minimum inhibitory drug concentrations, allowing for slowed but nonzero rates of proliferation. Previous work indicates that drug interactions may modulate evolution in different ways at higher drug concentrations [27]. In addition, our experiments were performed in planktonic populations, while many of the high-inoculum infections requiring combination treatment are likely to involve surface-associated biofilms, where spatial heterogeneity and complex community dynamics can dramatically alter the response to antibiotics. In *E. faecalis*, for example, population density can significantly modulate growth dynamics [8], while sub-inhibitory doses of cell wall inhibitors may actually promote biofilm growth [62]. Recent work in other bacterial species also shows that evolutionary adaptation may differ between biofilm and planktonic communities [63, 64]. Investigating adaptation to drug combinations in these different regimes, both at clinically-relevant concentrations and in biofilms, remains an interesting avenue for future work.

There are also several notable technical limitations. To minimize batch effects due to day-to-day variability (due to small changes in, for example, drug stock, media composition or temperature), we perform all experiments for a given drug pair together during the same 3-day period using the same reagent batches. We do see considerable variability between evolutionary replicates, even when they are performed side-by-side during the same period, with the same batch of media, and under the same selection conditions, giving us hope that these conditions capture at least some of the complexity of the possible evolutionary outcomes. However, it is possible that additional trajectories could arise, and perhaps even dominate, if

the experiments were repeated multiple times across different days or different growth conditions. In addition, it is clear that all growth curves are not purely exponential, and in fact the per capita growth rate can change with time. Our growth rate estimates should therefore be thought of as an effective growth rate that reduces the population dynamics each day to a single number. Similarly, the rescaling analysis assumes that the drug resistance in the population can be captured by a single $IC_{50}$ (for each drug), essentially neglecting clonal interference in favor of a single dominant resistant phenotype. In addition, the rescaling analysis does not incorporate fitness cost (which we did not measure). As a result of these limitations, the trends predicted by rescaling can only be evaluated qualitatively, though these limitations could potentially be overcome with significantly more experimental data, leading to more quantitative rescaling predictions. For example, resistance phenotyping of individual isolates from each population could provide insight into population heterogeneity during adaptation. Nevertheless, given the potential complexity of evolutionary trajectories, it is encouraging that simple rescaling arguments can qualitatively capture the coarse-grained features we measured. Future studies that aim to overcome the technical limitations of this work may be able to further evaluate quantitative agreement between specific evolutionary trajectories and the predictions of rescaling.

Most importantly, we stress that our results are based on in vitro laboratory experiments, which provide a well-controlled but potentially artificial–and certainly simplified–environment for evolutionary selection. While in vitro studies form the basis for many pharmacological regimens, the ultimate success or failure of new therapies must be evaluated using in vivo model systems and, ultimately, controlled clinical trials. We hope the results presented here offer a provocative look at evolution of *E. faecalis* in multidrug environments, but it is clear that these findings are not directly transferable to the clinic. Our results do include some clinically relevant antibiotic combinations, though we also sought a wide range of drug interaction types, including multiple antagonistic combinations that are unlikely a priori choices for clinical use. In addition, the relatively high levels of resistance of the ancestor strain to some drugs–such as aminoglycosides–would make them unlikely clinical choices to these populations. It is notable, however, that the resulting non-synergistic combinations often produced considerably slower growth adaptation, consistent with previous results that highlight potential benefits for non-standard combinations [40, 41, 43].

Finally, our results reveal that simple rescaling arguments—similar to those originally introduced in [41, 43]—can be used to understand many features of evolution in two-drug environments. Extending and formalizing these qualitative findings using stochastic models where fast evolutionary dynamics are coupled to geometric rescaling on adiabatically changing interaction landscapes is an exciting avenue for theoretical work that may provide both general insight as well as specific, experimentally testable predictions for how resistance evolves in multi-drug environments.

## Materials and methods

### Strains, media, and growth conditions

All experiments were performed on the V583 strain of *E. faecalis*, a fully sequenced clinical isolate. Overnight seed cultures were inoculated from a single colony and grown in sterilized brain heart infusion (BHI) medium at 37C with no shaking. Antibiotic stock solutions (Table 1) were prepared using sterilized Millipore water, diluted and aliquoted into single use micro-centrifuge tubes and stored at -20C or -80C. All drugs and media were purchased from Dot Scientific, Sigma-Aldrich or Fisher Scientific.

**Table 1. Antibiotics used in this study.**

| Antibiotic | Class | Mechanism |
|---|---|---|
| Ampicillin (AMP) | β-lactam (penicillins) | cell wall synthesis inhibitor |
| Ceftriaxone (CRO) | β-lactam (cephalosporin) | cell wall synthesis inhibitor |
| Ciprofloxacin (CIP) | fluoroquinolone | DNA synthesis inhibitor |
| Streptomycin (STR) | aminoglycoside | protein synthesis inhibitor |
| Tigecycline (TGC) | glycylcycline | protein synthesis inhibitor |

## Laboratory evolution and growth measurement

Evolution experiments were seeded by diluting overnight cultures of ancestral V583 cells 400x into individual wells of a 96-well microplate containing appropriate drug concentrations. All plates for a multi-day evolution experiment were prepared in advance by adding appropriate drug concentrations to 200 $\mu$L BHI and storing at -20C for not more than 4 days. Each day, a new plate was thawed and inoculated with 2 $\mu$L (100X dilution) from the previous day's culture. Plates were then sealed with BIO-RAD Microseal film to minimize evaporation and prevent cross contamination. Optical density at 600 nm (OD) was measured for each population every 20-25 minutes using an EnSpire Multimode Plate Reader with multi-plate stacker attachment located in a temperature-controlled (30C) warm room. Control wells containing ancestral cells and BHI medium were included on each plate as a growth control and for background subtraction, respectively. All dilutions and daily transfers were performed inside a ThermoFisher 1300 Series A2 safety cabinet to minimize contamination. Samples from each population were stocked in 15 percent glycerol and stored at -80C.

## Estimating per capita growth rate and drug response surfaces

We estimated per capita growth rate ($g$) from OD time series by fitting the early exponential phase portion of the background subtracted curves (0.1<OD<0.4) to an exponential function ($OD \sim e^{gt}$) using nonlinear least squares (MATLAB 7.6.0 curve fitting toolbox, Mathworks). The OD range [0.1, 0.4] was chosen because it spans a large region of early exponential phase growth in ancestor cells grown without drug. If the growth rate was estimated to be negative or the growth curve did not reach 0.1 over the course of the experiment, growth was set to 0. See S22 Fig for examples, including those where growth is clearly not exponential and therefore $g$ should be interpreted as an effective growth parameter, not the true per capita growth. We normalized all growth rates by the growth rate of ancestral cells in the absence of drugs performed on the same day, with one exception: ancestor growth curves for CRO-AMP and CRO-CIP were shared between the two experiments, which were performed on consecutive days. Ancestor growth rates varied slightly day-to-day, with a minimum of 0.74 and a maximum of 0.88 hr$^{-1}$ (doubling times of 47 and 56 mins, respectively). We visualized two-drug growth response surfaces by smoothing (2-d cubic spline interpolation) to reduce experimental noise and displaying smoothed surfaces as two-dimensional heat maps. When relevant–for example, for rescaling analysis–growth in unsampled regions of the growth surface was estimated with 2d interpolation.

## Phenotypic resistance profiling

Experiments to estimate the half-maximal inhibitory concentration (IC$_{50}$) for each population were performed in replicates of 3 in 96-well plates. Prior to IC$_{50}$ testing, frozen stocks for each population were swabbed and grown overnight in drug-free medium. These overnight cultures

were then diluted 100X into new plates containing fresh media and a gradient of 8 drug concentrations. We chose linearly spaced drug concentrations (rather than the more conventional 2-fold dilutions [65]) to increase precision of $IC_{50}$ estimates. After 20 hours of growth the optical density at 600 nm (OD600) was again measured and used to create a dose response curve. To quantify drug resistance, the resulting dose response curve was fit to a Hill-like function $f(x) = (1 + (x/K)^h)^{-1}$ using nonlinear least squares fitting, where $K$ is the half-maximal inhibitory concentration ($IC_{50}$) and $h$ is a Hill coefficient describing the steepness of the dose-response relationship. To reduce day-to-day fluctuations, control dose response curves (in replicates of 6) in ancestral cells were measured side-by-side each day with dose response curves of the adapted populations. The $IC_{50}$'s from these controls were used as the normalizing factor in calculating resistance relative to ancestor (i.e. $\log_2$-scaled fold change). See S7–S14 Figs for examples. Relative error (standard error of mean / mean) for $IC_{50}$ is typically on the order of ten percent and often much smaller (S15 Fig).

Estimated growth surfaces for ancestral cells as well as all estimated growth rates and $IC_{50}$ values are available in ref [66].

## Supporting information

**S1 Fig. Growth curves for adaptation in AMP-CRO combinations.** OD time series for populations grown in conditions A (top 2 rows, red), B (magenta), C (cyan), and D (last 2 rows, blue) for combinations of ceftriaxone (CRO) and ampicillin (AMP). Light gray curve is ancestor control in absence of drug. Days are green (0), blue (1), red (2), and magenta (3). Shaded region corresponds to OD range [0.1,0.4] over which growth rate is estimated.
(EPS)

**S2 Fig. Adaptation rate trends do not depend sensitively on choice of OD windows for growth rate fits.** Adaptation rates for populations grown in conditions A (red), B (magenta), C (cyan), and D (blue) for combinations of ceftriaxone (CRO) and ampicillin (AMP). Upper insets show the OD time series of ancestor strains in the absence of drug, with shaded region the OD range over which exponential growth rate is estimated (OD = [0.1,0.4], upper left; OD = [0.05,0.3], upper right; OD = [0.15,0.35], lower left; OD = [0.15,0.5], lower right). In all cases, adaptation rate in the single drug conditions (red and blue conditions, combined) is less than that in the combined drug conditions (magenta and cyan, combined), 1-sided t-test (unequal variance), $p < 0.05$.
(EPS)

**S3 Fig. Growth rate time series do not depend sensitively on choice of OD windows for growth rate fits.** Per capita growth rate over time for each condition, A (red), B (magenta), C (cyan), and D (blue), for combinations of ceftriaxone (CRO) and ampicillin (AMP). Rows 1-4 correspond to the four choices of OD windows in S2 Fig.
(EPS)

**S4 Fig. Growth rate time series and fits to linear and saturating functions.** Growth rate time series (circles) and both linear and saturating fits to determine mean adaptation rate (lines) for populations grown in conditions A (top 3 rows, red), B (magenta), C (cyan), and D (last 3 rows, blue) for combinations of ceftriaxone (CRO) and ampicillin (AMP). Saturating fits correspond to a functional form $y = y_{max} x/(x + K_{inv})$, where $y_{max}$ is the maximum possible growth rate (set to 1.2, slightly higher than 1 to allow for potential increases in growth relative to ancestor) and $K_{inv}$ is defined as the inverse adaptation rate.
(EPS)

**S5 Fig. Adaptation rate trends do not depend sensitively on linear or saturating fits to growth rate time series.** Adaptation rates for populations grown in conditions A (red), B (magenta), C (cyan), and D (blue) for combinations of ceftriaxone (CRO) and ampicillin (AMP) for linear fits (left) and saturating fits (right). Note that the vertical axis scales differ, as do the definition of adaptation rate. In the linear case, adaptation rate is in units of growth rate per unit time, while in the saturating case adaptation rate is in units of inverse time. In all cases, adaptation rate in the single drug conditions (red and blue conditions, combined) is less than that in the combined drug conditions (magenta and cyan, combined), 1-sided t-test (unequal variance), $p < 10^{-3}$.
(EPS)

**S6 Fig. Median growth curves within each condition capture qualitative trends of growth rate adaptation.** Median growth curves–which consist of the median OD across all populations within a given condition at each time point–for the four drug combinations on the final day of adaptation. Color scheme is same as in the main text: red and blue are single drug conditions, magenta and cyan correspond to drug combinations. Shaded region corresponds to OD range over which exponential growth rates are estimated in the main text.
(EPS)

**S7 Fig. Dose response curves to estimate AMP resistance following CRO adaptation.** Dose response curves for six populations ("mutants", rows) over 3 days (columns) for populations adapted to condition A (CRO-only). Dots are measurements (3 technical replicates), lines are estimated fits to Hill-like dose response function. Shaded region indicates mean $IC_{50} \pm 2$ standard errors across replicates. Light gray curves and x symbols correspond to ancestor strain control.
(EPS)

**S8 Fig. Dose response curves to estimate AMP resistance following AMP-CRO adaptation.** Dose response curves for six populations ("mutants", rows) over 3 days (columns) for populations adapted to condition B (AMP-CRO). Dots are measurements (3 technical replicates), lines are estimated fits to Hill-like dose response function. Shaded region indicates mean $IC_{50} \pm 2$ standard errors across replicates. Light gray curves and x symbols correspond to ancestor strain control.
(EPS)

**S9 Fig. Dose response curves to estimate AMP resistance following AMP-CRO adaptation.** Dose response curves for six populations ("mutants", rows) over 3 days (columns) for populations adapted to condition C (AMP-CRO). Dots are measurements (3 technical replicates), lines are estimated fits to Hill-like dose response function. Shaded region indicates mean $IC_{50} \pm 2$ standard errors across replicates. Light gray curves and x symbols correspond to ancestor strain control.
(EPS)

**S10 Fig. Dose response curves to estimate AMP resistance following AMP adaptation.** Dose response curves for six populations ("mutants", rows) over 3 days (columns) for populations adapted to condition D (AMP-only). Dots are measurements (3 technical replicates), lines are estimated fits to Hill-like dose response function. Shaded region indicates mean $IC_{50} \pm 2$ standard errors across replicates. Light gray curves and x symbols correspond to ancestor strain control.
(EPS)

**S11 Fig. Dose response curves to estimate CRO resistance following CRO adaptation.** Dose response curves for six populations ("mutants", rows) over 3 days (columns) for populations adapted to condition A (CRO-only). Dots are measurements (3 technical replicates), lines are estimated fits to Hill-like dose response function. Shaded region indicates mean $IC_{50} \pm 2$ standard errors across replicates. Light gray curves and x symbols correspond to ancestor strain control.
(EPS)

**S12 Fig. Dose response curves to estimate CRO resistance following AMP-CRO adaptation.** Dose response curves for six populations ("mutants", rows) over 3 days (columns) for populations adapted to condition B (AMP-CRO). Dots are measurements (3 technical replicates), lines are estimated fits to Hill-like dose response function. Shaded region indicates mean $IC_{50} \pm 2$ standard errors across replicates. Light gray curves and x symbols correspond to ancestor strain control.
(EPS)

**S13 Fig. Dose response curves to estimate CRO resistance following AMP-CRO adaptation.** Dose response curves for six populations ("mutants", rows) over 3 days (columns) for populations adapted to condition C (AMP-CRO). Dots are measurements (3 technical replicates), lines are estimated fits to Hill-like dose response function. Shaded region indicates mean $IC_{50} \pm 2$ standard errors across replicates. Light gray curves and x symbols correspond to ancestor strain control.
(EPS)

**S14 Fig. Dose response curves to estimate CRO resistance following AMP adaptation.** Dose response curves for six populations ("mutants", rows) over 3 days (columns) for populations adapted to condition A (AMP-only). Dots are measurements (3 technical replicates), lines are estimated fits to Hill-like dose response function. Shaded region indicates mean $IC_{50} \pm 2$ standard errors across replicates. Light gray curves and x symbols correspond to ancestor strain control.
(EPS)

**S15 Fig. Relative error in resistance measurements for AMP and CRO.** Histogram of relative error, defined as the standard error of the mean divided by the mean, for $IC_{50}$ estimates from 72 populations (6 examples per condition, 4 conditions, 3 days) per drug adapted to AMP-CRO conditions. In each case, standard error is calculated over three technical replicates.
(EPS)

**S16 Fig. Growth curves for adaptation in AMP-STR combinations.** OD time series for populations grown in conditions A (top 2 rows, red), B (magenta), C (cyan), and D (last 2 rows, blue) for combinations of ampicillin (AMP) and streptinomycin (STR). Light gray curve is ancestor control in absence of drug. Days are green (0), blue (1), and red (2). Shaded region corresponds to OD range [0.1,0.4] over which growth rate is estimated.
(EPS)

**S17 Fig. Growth rate time series for adaptation in AMP-STR combinations.** Growth rate time series (circles) and linear fits to determine mean adaptation rate (lines) for populations grown in conditions A (top 3 rows, red), B (magenta), C (cyan), and D (last 3 rows, blue) for combinations of ampicillin (AMP) and streptinomycin (STR).
(EPS)

**S18 Fig. Growth curves for adaptation in CRO-CIP combinations.** OD time series for populations grown in conditions A (top 2 rows, red), B (magenta), C (cyan), and D (last 2 rows, blue) for combinations of ceftriaxone (CRO) and ampicillin (AMP). Light gray curve is ancestor control in absence of drug. Days are green (0), blue (1), red (2), and magenta (3). Shaded region corresponds to OD range [0.1,0.4] over which growth rate is estimated. (EPS)

**S19 Fig. Growth rate time series for adaptation in CRO-CIP combinations.** Growth rate time series (circles) and linear fits to determine mean adaptation rate (lines) for populations grown in conditions A (top 2 rows, red), B (magenta), C (cyan), and D (last 2 rows, blue) for combinations of ceftriaxone (CRO) and ciprofloxacin (CIP). (EPS)

**S20 Fig. Growth curves for adaptation in TGC-CIP combinations.** OD time series for populations grown in conditions A-K (11 rows, 8 replicates per condition). Light gray curve is ancestor control in absence of drug. Days are green (0), blue (1), and red (2). Shaded region corresponds to OD range [0.1,0.4] over which growth rate is estimated. (EPS)

**S21 Fig. Growth rate time series for adaptation in TGC-CIP combinations.** Growth rate time series for each population (light lines) and the mean across populations for a given condition (A-K) for combinations of tigecycline (TGC) and ciprofloxacin (CIP). (EPS)

**S22 Fig. Example exponential fits to growth curves.** OD time series for populations grown in conditions A (top 2 rows), B (rows 3-4), C (rows 5-6), and D (last 2 rows) for combinations of ampicillin (AMP) and streptinomycin (STR). Light gray curve is ancestor control in absence of drug. Days are green (0), blue (1), and red (2). Lines show fits to exponential functions. Shaded region corresponds to OD range [0.1,0.4] over which growth rate is estimated. Note that growth rates less than zero (see several examples on top row) were set to 0 for all subsequent analysis. (EPS)

## Acknowledgments

We would like to thank Naji Kabbani and Zahra Arabzada for their technical support and contributions to the early stages of this project.

## Author Contributions

**Conceptualization:** Ziah Dean, Kevin B. Wood.

**Formal analysis:** Ziah Dean, Jeff Maltas, Kevin B. Wood.

**Funding acquisition:** Kevin B. Wood.

**Investigation:** Ziah Dean, Jeff Maltas.

**Methodology:** Ziah Dean, Jeff Maltas, Kevin B. Wood.

**Supervision:** Kevin B. Wood.

**Validation:** Kevin B. Wood.

**Visualization:** Ziah Dean, Kevin B. Wood.

**Writing – original draft:** Ziah Dean, Kevin B. Wood.

**Writing – review & editing:** Ziah Dean, Jeff Maltas, Kevin B. Wood.

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
