## [Decision Letter · Decision Letter 0]

5 Aug 2019

Dear Dr. Wood,

Thank you very much for submitting your manuscript "Antibiotic interactions shape short-term evolution of resistance in Enterococcus faecalis" (PPATHOGENS-D-19-01118) for review by PLOS Pathogens. Your manuscript was fully evaluated at the editorial level and by independent peer reviewers. The reviewers appreciated the attention to an important problem, but raised some substantial concerns about the manuscript as it currently stands. These issues must be addressed before we would be willing to consider a revised version of your study. We cannot, of course, promise publication at that time.

We therefore ask you to modify the manuscript according to the review recommendations before we can consider your manuscript for acceptance. Your revisions should address the specific points made by each reviewer.

(1) A letter containing a detailed list of your responses to the review comments and a description of the changes you have made in the manuscript. Please note while forming your response, if your article is accepted, you may have the opportunity to make the peer review history publicly available. The record will include editor decision letters (with reviews) and your responses to reviewer comments. If eligible, we will contact you to opt in or out.

(2) Two versions of the manuscript: one with either highlights or tracked changes denoting where the text has been changed; the other a clean version (uploaded as the manuscript file).

Additionally, to enhance the reproducibility of your results, PLOS recommends that you deposit your laboratory protocols in protocols.io, where a protocol can be assigned its own identifier (DOI) such that it can be cited independently in the future. For instructions see http://journals.plos.org/plospathogens/s/submission-guidelines#loc-materials-and-methods

We hope to receive your revised manuscript within 60 days. If you anticipate any delay in its return, we ask that you let us know the expected resubmission date by replying to this email. Revised manuscripts received beyond 60 days may require evaluation and peer review similar to that applied to newly submitted manuscripts.

[LINK]

Sincerely,

Michael Otto

Associate Editor

PLOS Pathogens

François Balloux

Section Editor

PLOS Pathogens

Kasturi Haldar

Editor-in-Chief

PLOS Pathogens

orcid.org/0000-0001-5065-158X

Grant McFadden

Editor-in-Chief

PLOS Pathogens

orcid.org/0000-0002-2556-3526

Reviewer's Responses to Questions

**Part I - Summary**

Reviewer #1: This is a potentially interesting study of resistance evolution to drug combinations in E. faecalis. The authors acquired growth curves of serially passaged E. faecalis in the presence of four different drug combinations, and within each combination examined each single drug and then 2 or more mixtures of the two drugs, all expected to have the same initial impact on growth rate. They conclude that adaptation trajectories are different for different dosing combinations and that this can be explained by rescaling the drug concentration to an effective concentration, by using final resistant profiles, and predicting growth rate from ancestral two drug growth response surface.

Reviewer #2: Dean et al present a compelling study of how different antibiotic combination treatments affect the subsequent phenotypes evolved under such treatment. The choice of combinations is clever, demonstrating examples of synergistic and antagonistic stresses between the antibiotics. Most interesting is the tagacycline/ciprofloxacin combination that has a critical value of tagacycline concentration when there is a sudden loss of cipro resistance. Subsequently, they come up with a rescaling to normalize the various cultures to simulated constant growth. The result is quite striking: bi-antibiotic resistance profiles are essentially linear. This is a wonderful result, showing that the phenomenological result of experimental evolution follows a quite simple rule!

I have the following comments:

p. 6, Fig 1c: I am trying to understand how the adaptation rate can differ but the resistance is almost identical between conditions. The authors note that this is a surprising result, but I would appreciate slightly more exposition on why I should not feel so naive being surprised about it.

p. 6, note on intro paragraph to section titled: “Aminoglycoside/β-lactam and β-lactam/fluoroquinolone combinations slow growth adaptation and select for resistant profiles distinct from those evolved to the component drugs.” - some may note that in most papers this section would belong in the introduction, but I believe that it is well-suited here because there are multiple antibiotic combinations done, so this reminds the reader of the importance of this particular combination.

Rescaling - I personally find this to be compelling and interesting, but I fear that some in the pathogens community will not appreciate how important it is, based on my prior experience. My hope is that somewhat challenging concepts from other fields can be allowed to stand.

Reviewer #3: In this manuscript the authors study the effect of drug combinations on the evolution of resistance. They use test cases of different drug interactions to show how it drives the resistance evolution. They use the picture of geometric rescaling to explain the results of the evolution experiments. To my knowledge, this kind of detailed experiments and analysis is innovative and can help with the optimization problem of effectiveness vs resistance. Following some revisions/clarifications, I think the manuscript will be more than adequate for publication.

**Part II – Major Issues: Key Experiments Required for Acceptance**

Reviewer #1: 1) A critical missing piece to the manuscript is a description of how exactly the per capita growth rate is measured. Almost all results and analyses depend critically on this value, and there are several places where results appear to disagree with each other

In the methods it is stated that the early exponential phase was fit to an exponential function and then normalized to ancestral cells

- The growth curves of those ancestral cells are needed

Are all data normalized to one set of controls? Are they normalized to matched controls done alongside each experiment?

Why in Fig 1 and 2 do the evolved populations struggle to meet a growth rate of ‘1’, despite acquiring resistance, whereas in Fig 3 combination D exceeds 1 – did this condition really evolve an E. faecalis strain that grows faster than wildtype?

- A more precise description than “typically OD<0.4” is needed to describe how the early exponential phase of the growth curve was defined and how lag phase was avoided

Presumably the growth rate is related to the slope of the exponential portion of the OD600 over time curve. If so:

Fig 1a condition D the D0 and D3 slopes look identical, and yet the ‘growth rate’ is very different when plotted in Fig1b.

Fig 2a condition A, the D0 slope looks steeper than the D2 slope, and yet the ‘growth rate’ in Fig 2b is close to 0 for D0 and at 0.5 for D2

2) It is difficult to assess if the differences are real and significant

-Each drug combination appears to have been examined in one serial passaging experiment, sometimes for two days, sometimes for three days. Would the trajectories and resistance evolution be reproducible in a second experiment?

-the adaptation rate is measured by a linear fit to very non-linear data. For many conditions it seems clear that the growth rate increases and then plateaus, so the linear fit reduces the adaptation rate. Would a better measure be time to maximum growth rate?

-Resistance is reported as the change in IC50, and often the level of resistance is fairly low (a two to three fold change), and several conclusions rest on “differences” in the time to resistance or on the degree of resistance acquired. However; it seems a typical MIC assay was used to assess resistance where cultures are grown overnight in presence of drug and then OD600 measured. This usually leads to stationary cultures below the inhibitory concentrations, and therefore a steep delineation between growth and no growth, with often 2-fold differences in MIC values for day to day assay variation for a given strain and antibiotic. This makes me skeptical that small observed changes in the IC50 are significant. The dose-response curves used to calculate IC50 could also be given as supplemental figures.

Perhaps a better dynamic range and more reliable IC50 would be obtained by measuring the growth rate of the evolved populations to a series of dilutions of the single drugs, this measure might also better align with the other data.

- The combination where these issues are most clearly seen is CRO-AMP. The authors state that for the experiment in Fig 1 “the day 2 curves vary substantially between replicates and conditions”. The day 3 curves in Fig 1a all look fairly similar to my eye, especially when compared to the other figures in the paper. Secondly they state that the “adaptation is significantly faster for the two combinations (B and C) than for the single drug treatments (A and B)”. Was a test used to assess significance? If so which one?

Would the data be reproducible if the experiment was repeated again on a different day?

In Fig 3, condition A is the same as in Fig 1, and the growth rate curve in panel b is substantially different between days (In Fig 1 it plateaus at 0.5-0.6 on day 1, in Fig 2 it plateaus at 1 on day 2). The overall adaptation rate is similar, both close to 0.2, and so is the level of resistance measured; however the difference between the adaptation rate between experiments for condition A in Fig 1 and 3 looks similar to the difference in adaptation rate within experiment between conditions A and C in Fig 1.

3) A major conclusion of the paper is that rescaling concentrations based on resistance profiles predicts the growth rate in the presence of the combination. This is an attractive conclusion, most robust for the TGC-CIP combination. In this case it looks as though the predicted growth rate actually maps very closely to the achieved growth rates in Fig S4. Would be useful to plot the predicted versus actual to drive this point home?

However based on points listed above and below the specificity of predictions and conclusions is hard to agree with for several of the other combinations

- For combination CRO-AMP why is the predicted growth rate in Fig 5 a better than 1? Is a difference between a growth rate of 1 and 1.1 really real and significant? My conclusion would be much more general that concentration rescaling predicts in all of the combinations the adapted populations should have near wildtype growth, and that matches what was observed.

- For AMP-STR and CRO-CIP the authors claim the resistant profiles fall on a line segment.

For AMP-STR the resistant changes are so small that it is hard to agree with this and for CRO-CIP it actually looks more like a curve, which is what I would expect from the curved antagonistic growth contours in Fig 3 a.

The line segment analysis is unsatisfying. What was used to determine the end points X and Y of the line segments? Why do they not lie at the x and y intercepts of the dotted lines through 0 resistance, or why do they not lie at the points of A and D, the single drug conditions?

Reviewer #2: No new experiments required.

Reviewer #3: To my opinion there are no new experiments required, but part the analysis must be revised/explained.

Looking at Figure 1a (right) the OD curves (D3) of A and D conditions looks different (it looks like in condition A the adaptation is faster), but A and D has the same adaptation rate in Figure 1b.

Similar issue is with conditions A and D in Figure 2 - it looks like the adaptation is faster condition D, but the adaptation rate are equal in Figure 2b. Moreover, it looks like the growth on day zero are not equal for all conditions.

I guess this due to the criterion for the time interval of "early exponential phase". You mentioned that it is usually OD<0.4, please elaborate more on the exact definition and the its motivation. How does the results of the paper depend on the definition? Does the main results of the paper hold with a different definition?

You mentioned this issue as a limitation: "It is clear that all growth curves are not purely exponential, and in fact the per capita growth rate can change with time. Our growth rate estimates should therefore be thought of as an effective growth rate that reduces the population dynamics each day to a single number. "

It is great that you are aware of this, but the current figures are confusing for the reader because of this issue. Additionally to a more detailed explanation of this issue in the methods section, please add the time interval that was used to estimate the growth rate in the OD figures (figure 1a,2a,...).

**Part III – Minor Issues: Editorial and Data Presentation Modifications**

Reviewer #1: 4) There are several mistakes in the manuscript, pg 3 bottom line the authors refer to day 2 curves in Fig 1 but in Fig1a the curves are labeled D3. In Fig 1b the average rate of growth adaptation point for D has tiny error bars on it, but I do not see error bars on any other such point. On Pg 5 the authors refer to single-drug conditions (A and B), but presumably mean A and D, and this is repeated on pg 6

5) Two of the antagonistic combinations examined, AMP-STR, and CRO-CIP, include drugs that the E. faecalis strain used is already resistant to. This makes the concentrations used to study streptomycin and ceftriaxone relatively high, and lessens the relevance of the study

6) In Fig 2 parts a and b show results up to day 2, but part c resistance profiles are shown up to day 3. The data should be consistent

7) In the discussion the authors speculate that OD measurements at shorter intervals would allow better growth rate estimates – it is hard to imagine that acquiring more frequently than once in 20 minutes would make a drastic difference.

8) The growth rate curve in Fig 1b for condition D is surprising given the population is already resistant to AMP by day 1 according to Fig 1c. I suspect something went awry experimentally or with the analysis of day 1 growth rate.

Reviewer #2: See summary in Part I.

Reviewer #3: In Figure 1a, the legend is D3, in the text you mentioned day 2.

Are the OD curves in log scale? please give some indication on the figure.

Why the D0 curves doesn't start at t=0?

PLOS authors have the option to publish the peer review history of their article (what does this mean?). If published, this will include your full peer review and any attached files.

Reviewer #1: No

Reviewer #2: Yes: Christian Ray

Reviewer #3: No

---

## [Decision Letter · Decision Letter 1]

11 Dec 2019

Dear Dr. Wood,

We are pleased to inform that your manuscript, "Antibiotic interactions shape short-term evolution of resistance in Enterococcus faecalis", has been editorially accepted for publication at PLOS Pathogens. 

Before your manuscript can be formally accepted and sent to production, you will need to complete our formatting changes, which you will receive by email within a week. Please note that your manuscript will not be scheduled for publication until you have made the required changes.

IMPORTANT NOTES

(1) Please note, once your paper is accepted, an uncorrected proof of your manuscript will be published online ahead of the final version, unless you’ve already opted out via the online submission form. If, for any reason, you do not want an earlier version of your manuscript published online or are unsure if you have already indicated as such, please let the journal staff know immediately at plospathogens@plos.org.

(2) Copyediting and Proofreading: The corresponding author will receive a typeset proof for review, to ensure errors have not been introduced during production. Please review the PDF proof of your manuscript carefully, as this is the last chance to correct any errors. Please note that major changes, or those which affect the scientific understanding of the work, will likely cause delays to the publication date of your manuscript. 

(3) Appropriate Figure Files: Please remove all name and figure # text from your figure files. Please also take this time to check that your figures are of high resolution, which will improve the readbility of your figures and help expedite your manuscript's publication. Please note that figures must have been originally created at 300dpi or higher. Do not manually increase the resolution of your files. For instructions on how to properly obtain high quality images, please review our Figure Guidelines, with examples at: http://journals.plos.org/plospathogens/s/figures.

(4) Striking Image: Please upload a striking still image to accompany your article if one is available (you can include a new image or an existing one from within your manuscript). Should your paper be accepted, this image will be considered for our monthly issue image and may also appear on our website to feature your article. Please upload this as a separate file, selecting "striking image" as the file type upon upload. Please also include a separate "Other" file with a caption, including credits and any potential copyright information. Please do not include the caption in the main article file. If your image is from someone other than yourself, please ensure that the artist has read and agreed to the terms and conditions of the Creative Commons Attribution License at http://journals.plos.org/plospathogens/s/content-license. Please note that PLOS cannot publish copyrighted images.

(5) Press Release or Related Media: If your institution or institutions have a press office, please notify them about your upcoming paper at this point, to enable them to help maximize its impact. If they will be preparing press materials for this manuscript, please inform our press team in advance at plospathogens@plos.org as soon as possible. We ask that you contact us within one week to plan ahead of our fast Production schedule. If you need to know your paper's publication date for related media purposes, you must coordinate with our press team, and your manuscript will remain under a strict press embargo until the publication date and time. This means an early version of your manuscript will not be published ahead of your final version. 

(6)  PLOS requires an ORCID iD for all corresponding authors on papers submitted after December 6th, 2016. Please ensure that you have an ORCID iD and that it is validated in Editorial Manager.  To do this, go to ‘Update my Information’ (in the upper left-hand corner of the main menu), and click on the Fetch/Validate link next to the ORCID field.  This will take you to the ORCID site and allow you to create a new iD or authenticate a pre-existing iD in Editorial Manager

(7) Update your Profile Information: Now that your manuscript has been provisionally accepted, please log into Editorial Manager and update your profile, if needed. Go to https://www.editorialmanager.com/ppathogens, log in, and click on the "Update My Information" link at the top of the page. Please update your user information to ensure an efficient production and billing process. 

(8) LaTeX users only: Our staff will ask you to upload a TEX file in addition to the PDF before the paper can be sent to typesetting, so please carefully review our Latex Guidelines http://journals.plos.org/plospathogens/s/latex in the meantime.

(9) If you have associated protocols in protocols.io, please ensure that you make them public before publication to guarantee immediate access to the methodological details.

Best regards,

Michael Otto

Section Editor

PLOS Pathogens

François Balloux

Section Editor

PLOS Pathogens

Kasturi Haldar

Editor-in-Chief

PLOS Pathogens

orcid.org/0000-0001-5065-158X

Grant McFadden

Editor-in-Chief

PLOS Pathogens

orcid.org/0000-0002-2556-3526

Reviewer Comments (if any, and for reference):

Reviewer's Responses to Questions

**Part I - Summary**

Reviewer #1: This is a revised and improved version of the manuscript on the adaptation of E. faecalis to four different combinations of antibiotics. Their conclusions remain that within a drug pair, adaptation trajectories differ depending on the dosing combination of the component drugs, and that this can be explained by using resistance profiles to rescale concentrations. The authors have changed and performed several extra analyses, corrected errors, and provided more information. These changes improve the manuscript. The more interesting part of the study lies in the results and discussion on how resistance measures used to rescale the growth-response curve of a particular combination correlate with experimental outcome.

Method for calculating that growth-rate response is still not really clear and very difficult to understant. After considerable puzzling over the figures and methods, I now take it that essentially a straight line was fitted through the OD curves (ie exponential function with the y-axis on a log scale) in the range of OD 0.1 – 0.4 across the whole time scale. This combines the aspects of exponential growth rate and saturation density into one number to approximate the inhibitory effect of the drugs on the population.

There are still several issues that are confusing:

1) Why do all the growth rates on day 0 start essentially at 0?

The extent of inhibition from the drug conditions on D0 vary between the four experiments, and yet all of them seem to have a growth rate very near 0 in the growth rate over time curves in the part b of each figure.

To illustrate this, Fig 2, condition A, Amp alone. In Fig 2a, the growth curve for D0 seems to increase rapidly to OD 0.4 then plateau – surely even with the plateau blunting the “growth rate” or slope of the exponential fit, and even with normalizing to ancestral growth rate, the growth rate should still be above 0?

Also in Fig 1a, the location of the circles in the heatmap indicate that a contour of constant inhibition of approximately 0.5 was chosen, however, in part b the growth rates of D0 are all close to 0.

2) The growth curves in Fig S16 indicate that in Fig 2 for condition A the growth rate should decrease from Day 0 to Day 1, but this is not reflected in the graphs in panel b.

3) The authors have very helpfully included Fig S2 and Fig S3, examining if changing the OD window for calculating growth rate impacts the conclusions of the CRO-AMP combination. However, it does not seem that the differences between single drugs and drug combinations remain significant with the changed OD window. Specifically, the

legend of Fig S2 states that in all four methods a t.test shows significance comparing single drug conditions to drug combinations. For the panels on the right, examining OD [0.05,0.3] and OD [0.15,0.5] do not seem different.

Reviewer #2: The authors have gone to great lengths to address the first round of reviews. The strengths of the study largely remain intact, while at the same time the authors laudably discuss the weaknesses and limitations in scope of the study design. The combination of experimental evolution and sensitivity profiles to multiple drugs is significant both as a basic study of pathogen evolution that may emulate some in-host conditions and as a conceptual approach to see how multiple selective pressures change bacterial phenotypes.

Reviewer #3: In this manuscript, the authors study the effect of drug combinations on the evolution of

resistance. They use test cases of different drug interactions to show how it drives the resistance evolution. They use the picture of geometric rescaling to explain the results of the evolution experiments. To my knowledge, this kind of detailed experiments and analysis is innovative and can help with the optimization problem of effectiveness vs resistance. The author made a significant improvement in the MS but one issue is still unclear to me.

**Part II – Major Issues: Key Experiments Required for Acceptance**

Reviewer #1: This is a revised and improved version of the manuscript on the adaptation of E. faecalis to four different combinations of antibiotics. Their conclusions remain that within a drug pair, adaptation trajectories differ depending on the dosing combination of the component drugs, and that this can be explained by using resistance profiles to rescale concentrations. The authors have changed and performed several extra analyses, corrected errors, and provided more information. These changes improve the manuscript. The more interesting part of the study lies in the results and discussion on how resistance measures used to rescale the growth-response curve of a particular combination correlate with experimental outcome.

Method for calculating that growth-rate response is still not really clear and very difficult to understant. After considerable puzzling over the figures and methods, I now take it that essentially a straight line was fitted through the OD curves (ie exponential function with the y-axis on a log scale) in the range of OD 0.1 – 0.4 across the whole time scale. This combines the aspects of exponential growth rate and saturation density into one number to approximate the inhibitory effect of the drugs on the population.

There are still several issues that are confusing:

1) Why do all the growth rates on day 0 start essentially at 0?

The extent of inhibition from the drug conditions on D0 vary between the four experiments, and yet all of them seem to have a growth rate very near 0 in the growth rate over time curves in the part b of each figure.

To illustrate this, Fig 2, condition A, Amp alone. In Fig 2a, the growth curve for D0 seems to increase rapidly to OD 0.4 then plateau – surely even with the plateau blunting the “growth rate” or slope of the exponential fit, and even with normalizing to ancestral growth rate, the growth rate should still be above 0?

Also in Fig 1a, the location of the circles in the heatmap indicate that a contour of constant inhibition of approximately 0.5 was chosen, however, in part b the growth rates of D0 are all close to 0.

2) The growth curves in Fig S16 indicate that in Fig 2 for condition A the growth rate should decrease from Day 0 to Day 1, but this is not reflected in the graphs in panel b.

3) The authors have very helpfully included Fig S2 and Fig S3, examining if changing the OD window for calculating growth rate impacts the conclusions of the CRO-AMP combination. However, it does not seem that the differences between single drugs and drug combinations remain significant with the changed OD window. Specifically, the

legend of Fig S2 states that in all four methods a t.test shows significance comparing single drug conditions to drug combinations. For the panels on the right, examining OD [0.05,0.3] and OD [0.15,0.5] do not seem different.

Reviewer #2: None at this stage.

Reviewer #3: (No Response)

**Part III – Minor Issues: Editorial and Data Presentation Modifications**

Reviewer #1: This is a revised and improved version of the manuscript on the adaptation of E. faecalis to four different combinations of antibiotics. Their conclusions remain that within a drug pair, adaptation trajectories differ depending on the dosing combination of the component drugs, and that this can be explained by using resistance profiles to rescale concentrations. The authors have changed and performed several extra analyses, corrected errors, and provided more information. These changes improve the manuscript. The more interesting part of the study lies in the results and discussion on how resistance measures used to rescale the growth-response curve of a particular combination correlate with experimental outcome.

Method for calculating that growth-rate response is still not really clear and very difficult to understant. After considerable puzzling over the figures and methods, I now take it that essentially a straight line was fitted through the OD curves (ie exponential function with the y-axis on a log scale) in the range of OD 0.1 – 0.4 across the whole time scale. This combines the aspects of exponential growth rate and saturation density into one number to approximate the inhibitory effect of the drugs on the population.

There are still several issues that are confusing:

1) Why do all the growth rates on day 0 start essentially at 0?

The extent of inhibition from the drug conditions on D0 vary between the four experiments, and yet all of them seem to have a growth rate very near 0 in the growth rate over time curves in the part b of each figure.

To illustrate this, Fig 2, condition A, Amp alone. In Fig 2a, the growth curve for D0 seems to increase rapidly to OD 0.4 then plateau – surely even with the plateau blunting the “growth rate” or slope of the exponential fit, and even with normalizing to ancestral growth rate, the growth rate should still be above 0?

Also in Fig 1a, the location of the circles in the heatmap indicate that a contour of constant inhibition of approximately 0.5 was chosen, however, in part b the growth rates of D0 are all close to 0.

2) The growth curves in Fig S16 indicate that in Fig 2 for condition A the growth rate should decrease from Day 0 to Day 1, but this is not reflected in the graphs in panel b.

3) The authors have very helpfully included Fig S2 and Fig S3, examining if changing the OD window for calculating growth rate impacts the conclusions of the CRO-AMP combination. However, it does not seem that the differences between single drugs and drug combinations remain significant with the changed OD window. Specifically, the

legend of Fig S2 states that in all four methods a t.test shows significance comparing single drug conditions to drug combinations. For the panels on the right, examining OD [0.05,0.3] and OD [0.15,0.5] do not seem different.

Reviewer #2: None.

Reviewer #3: I still do not understand how did the authors evaluated the growth rate in a particular case. They mentioned OD in the range 0.1-0.4. I don’t understand what is the time interval for the fit in the case that the OD does not reach 0.4, is it 10 hours? What about the cases in which the OD reaches value larger than 0.4 and then declines? I am not sure that the definition of 0.1 to 0.4 is a good definition, because in the case of reaching saturation before 0.4 the effective growth rate depends on the time interval after saturation. A better definition could be the slope in the range where the log(OD) is linear. I leave it to author to decide whether to change the definition, but the clarification of the time interval (not just OD interval) is needed. Please provide the fit to exponent for some cases in the SI.

PLOS authors have the option to publish the peer review history of their article (what does this mean?). If published, this will include your full peer review and any attached files.

Reviewer #1: No

Reviewer #2: No

Reviewer #3: No

---

## [Editor Report · Acceptance letter]

20 Feb 2020

Dear Dr. Wood,

We are delighted to inform you that your manuscript, "Antibiotic interactions shape short-term evolution of resistance in Enterococcus faecalis," has been formally accepted for publication in PLOS Pathogens.

Best regards,

Kasturi Haldar

Editor-in-Chief

PLOS Pathogens

orcid.org/0000-0001-5065-158X

Michael Malim

Editor-in-Chief

PLOS Pathogens

orcid.org/0000-0002-7699-2064